# Synthesis of Xylan-Click-Quaternized Chitosan via Click Chemistry and Its Application in the Preparation of Nanometal Materials

**DOI:** 10.3390/molecules27113455

**Published:** 2022-05-27

**Authors:** Yuqiong Luo, Jihai Cai, Yanan Huang, Jiwen Luo

**Affiliations:** 1State Key Laboratory of Pulp & Paper Engineering, South China University of Technology, Guangzhou 510640, China; luo2020start@henu.edu.cn (Y.L.); jihaicai@scut.edu.cn (J.C.); 2Key Lab for Special Functional Materials of Ministry of Education, School of Materials, Henan University, Kaifeng 475004, China; 3Guangdong Provincial Key Laboratory of Chemical Pollution and Environmental Safety & MOE Key, Laboratory of Theoretical Chemistry of Environment, South China Normal University, Guangzhou 510640, China; 2021024401@m.scnu.edu.cn

**Keywords:** xylan, quaternized chitosan, click reaction, AgNPs, AuNPs

## Abstract

For the high-valued utilization of hemicelluloses and for realizing the controllable synthesis of NPs, this paper’s aim is to combine xylan, chitosan and nanometal materials at the same time. In this research study, firstly, propargyl xylan was synthesized via nucleophilic substitution reaction between xylan and propargyl bromide in NaOH solution. On the other hand, a tosyl group was introduced onto the 6th position of synthesized quaternized chitosan (QCS), and the azide group replaced the tosyl group to obtain 6-amido-QCS (QCS-N_3_). The synthesis conditions of the above reactions were optimized. Subsequently, the novel xylan-click-QCS polymer was obtained via click reaction between terminal alkyne groups on the xylan chains and azide groups on QCS. Then, AgNPs and AuNPs were synthesized by adopting the xylan-click-QCS polymer as the reducing and stabilizing agent, and the reaction conditions were optimized to obtain well-dispersed and highly stable nanoparticles. There were two kinds of Ag nanomaterials, with diameters of 10~20 nm and 2~5 nm, respectively, indicating the formation of Ag nanoclusters, except for Ag nanoparticles, in this reaction. The diameter of the synthesized AuNPs was 20~30 nm, which possessed a more uniform size distribution. The Ag nanoclusters with a smaller size (2~5 nm) could inhibit MCF-7 cell proliferation effectively, indicating their application potential in cancer therapy. The study gives a new approach to the high-value utilization of biopolymers.

## 1. Introduction

Noble-metal nanoparticles (NPs) have been widely used in various fields, such as catalysis [1,2], biosensors [3,4], antibacterial materials [5,6,7] and pharmaceuticals [8,9], due to their unique physical and chemical properties. Considering its wide application in biomedical and clinical practice, the green synthesis of NPs by adopting reducing sugars as reductants and stabilizers has attracted the attention of many [10,11,12,13]. Hemicelluloses are abundant and cheap natural polysaccharides with great potential application value [14]. The presence of hydroxyl groups (–OH) and carboxyl groups (–COOH) makes them candidates for the green synthesis of NPs. For example, green synthesized NPs were obtained by using plant extract containing hemicelluloses [15,16,17]. In our previous study, we prepared NPs in the presence of xylan [18,19], and the synthesized NPs with controllable shape were distributed uniformly after optimizing the reaction conditions. However, there were still some problems in those reactions; first, due to the relatively weak interaction of hydroxyl oxygen and NPs, excessive reducing sugars were added into the reaction system to stabilize the synthesized NPs, which may have affected the further application of the NPs. Second, the metal ions easily complexed with the reducing aldehyde groups, and the NPs synthesized in a short time (few minutes) with an uneven size distribution. The researchers, then, extended the reaction time to homogenize the size distribution, which made the reactions hard to control. Therefore, it is necessary to modify the reducing sugars to control the reaction rate and effectively stabilizing the synthesized NPs. Chitosan derivatives have been used to prepare noble-metal NPs. As a biocompatible polysaccharide with long molecular length, it forms a net structure packing the Ag NPs and effectively stabilizing the NPs [20,21]. Moreover, its biocompatibility extends its application in biomedicine, food chemistry, biosensors, etc. [22]. Hence, the xylan–chitosan complex may provide a more effective and controllable way for the green synthesis of NPs.

Nevertheless, hemicelluloses can be soluble in alkaline solution, while chitosan can only be soluble in acidic solution; most hemicelluloses–chitosan complexes have been prepared by aggregation or physical crosslinking rather than real chemical bonds [23,24,25,26,27], which may limit their homogeneous dispersion and stability. Proposed by Sharpless in the past decade, click chemistry is a class of chemical reactions well-known because of its full advantages of being effective, reliable and selective and its high yields with little or no byproducts [28,29,30]. One of the most popular reactions within the click chemistry philosophy is the Cu(I)-catalyzed Huisgen 1,3-dipolar cycloaddition reaction between azide and terminal alkyne groups, which has received increasing attention because of its simple and powerful modular synthesis approach [30,31,32]. Moreover, there have been some reports about the combination of the click reaction and NPs for further applications [33,34,35]. For example, Dondi et al. [36] synthesized highly size- and shape-controlled AgNPs via sugar triazole templates, and found that sugar triazole ligands can control the nucleation and growth phases of the synthesis and passivate the surface of the resultant AgNPs. Therefore, we speculated that the hemicelluloses–chitosan click reaction product may overcome the disadvantages of reducing sugars for NP synthesizing.

In this article, xylan—the model compound of hemicelluloses—was used as raw material; a water-soluble chitosan derivative—quaternized chitosan (QCS) was prepared under microwave irradiation in aqueous solution. Then, terminal propargyl-functionalized xylan and the azido-functionalized QCS (QCS-N_3_) were synthesized. Subsequently, the novel xylan-click-QCS polymer was created via the one-step alkyne–azide click reaction in the presence of Cu(I) (Appendix A). We also optimized the reaction conditions. The successful binding of xylan and QCS was confirmed and characterized by FT-IR and ^13^C NMR. Then, AgNPs and AuNPs were synthesized by adopting the xylan-click-QCS polymer as the reducing and stabilizing agent, and the reaction conditions were optimized. Finally, the anticancer property of the Ag nanomaterials was evaluated (Figure 1).

## 2. Results and Discussion

### 2.1. Synthesis and Characterization of Xylan-Click-QCS

#### 2.1.1. Synthesis of Propargyl Xylan (Pg-Xylan) and Azide-6-Quaternized Chitosan (QCS-N_3_)

Xylan was propargylated with propargyl bromide in this paper. As shown in Table 1, the DS of Pg-xylan first increased and then decreased with the prolonging of the reaction time, and the maximum DS was 0.23 at 50 min. A longer reaction time benefitted the DS of Pg-xylan; however, the DS decreased slightly with the increase in reaction time after 50 min. A higher DS was obtained when the reaction temperature increased from 30 to 50 °C, but a further rise in temperature to 60 °C led to a fall in the DS from 0.65 to 0.44. This may have been induced by the degradation of xylan at a high temperature, which led to the decrease in the DS. This result also suggests that the preparation of Pg-xylan should be controlled at a mild temperature. Moreover, an increase in the ratio of PgBr/xylan from 0.5:1 to 2:1 resulted in an increase in the DS of Pg-xylan from 0.07 to 0.65; thereafter, the DS increased slightly to 0.70. It shows that a high amount of PgBr has a positive influence on the substitution of hydroxyl groups on the xylan chains. The reason could be that there was more PgBr reagent in contact with xylan molecules at a higher concentration. Moreover, the influence of the ratio of PgBr/xylan from 2:1 to 3:1 was negligible. As a result, the optimized Pg-xylan with a DS value of 0.65 could be obtained under the condition of a PgBr/xylan ratio of 2:1 at 50 °C for 50 min under microwave irradiation.

As shown in Table 2, we also investigated the influence of the reaction conditions on the DS of QCS-N_3_ under microwave irradiation. Increasing the reaction temperature from 50 to 80 °C resulted in an increase in the DS from 0.42 to 1.26, but a further rise in temperature to 90 and 100 °C had a slight influence on the DS. Moreover, the DS of QCS-N_3_ exceeded 1 when the reaction temperature reached 80 °C, indicating that other than the 6th position, the 3rd position of QCS was substituted by the azide group in this reaction. Furthermore, the DS of QCS-N_3_ increased from 1.26 to 1.57 with the increase in the reaction time from 60 to 80 min, after which the DS decreased to 1.22 and 0.88 with a reaction time of 100 and 120 min. The reason for this decrease is probably the partial degradation of QCS-6-OTs over a prolonged reaction time. Moreover, the DS of QCS-N_3_ increased from 0.70 to 1.55 with the increase in the ratio of NaN_3_/QCS-6-OTs from 2:1 to 4:1. However, a higher NaN_3_ concentration showed little effect on the DS of QCS-N_3_, which gave a slight increase in value from 1.55 to 1.61. Thus, the optimized QCS-N_3_ with the DS value of 1.55 can be obtained under the conditions of a NaN_3_/QCS-6-OTs ratio of 4:1 at 80 °C for 80 min under microwave irradiation.

Figure 2a shows the FT-IR spectra of original xylan and Pg-xylan; clearly, the characteristic peaks at 3421, 2931, 1417, 1040 and 899 cm^−1^ originated from pure xylan. Compared with pure xylan, similar absorbances could be observed in the spectrum of Pg-xylan, indicating that the backbone structure of xylan remained after the reaction. Moreover, the characteristic peaks at 1417, 1040 and 899 cm^−1^ weakened after the reaction, which may have been induced by the effect of microwave irradiation [18]. On the other hand, a new adsorption band appeared at 2117 cm^−1^, corresponding to C≡C stretching, indicating the successful incorporation of terminal alkyne groups onto the xylan chains [37,38].

Figure 2b shows the FT-IR spectra of chitosan, QCS, QCS-6-OTs and QCS-N_3_. In the spectrum of chitosan, the absorption peak at 3480 cm^−1^ was ascribed to O-H stretching combined with N-H stretching vibration, while the characteristic absorption band at 1592 cm^−1^ was assigned to the vibration of −NH_2_ [39]. Compared with chitosan, the characteristic peak (1592 cm^−1^) of -NH_2_ was weakened in the spectrum of QCS, and a new peak at 1483 cm^−1^ appeared, attributable to the methyl groups in quaternary ammonium salt. The results prove that the chemical modification of chitosan with ETA resulted in the *N*-monosubstitution of chitosan, which coincides with the previous study [40]. In the spectrum of QCS-6-OTs, there were characteristic peaks of the tosyl group at 1173 cm^−1^ and 900~600 cm^−1^, indicating the success of the reaction between the tosyl group and QCS. In the spectrum of QCS-6-OTs, the characteristic peaks at 1173 cm^−1^ and 900~600 cm^−1^ were attributable to the disappearance of the tosyl group, indicating that the tosyl group was substituted by other groups. Noticeably, the presence of the new band between 2200 and 2000 cm^−1^ was assigned to -N_3_ stretching. These new bands confirmed the successful introduction of azide groups onto the QCS backbone [41].

#### 2.1.2. Synthesis of Xylan-Click-QCS Polymer via Click Chemistry

Figure 3 shows the FT-IR spectra of Pg-xylan, QCS-N_3_ and xylan-click-QCS. After the click reaction, the band at 2117 cm^−1^ due to C≡C stretching and the bands at 2131 and 2038 cm^−1^ due to -N_3_ stretching completely disappeared, and the characteristic band of C=C appeared at 775 cm^−1^. Moreover, the characteristic absorbance bands of xylan and QCS still appeared in the spectrum of xylan-click-QCS, verifying that the reaction between alkyne groups on xylan and azide groups on QCS took place via click chemistry [42].

Figure 4 shows the ^13^C NMR spectra of xylan and Pg-xylan. In the spectrum of xylan, five major signals at δ 101.53 (C-1), 75.90 (C-4), 73.70 (C-3), 72.56 (C-2) and 62.78 (C-5) ppm corresponded to (1→4)-β-D-xylopyranosyl units. Other small signals in the region 50–110 ppm were attributed to arabinofuranosyl and 4-O-methy-α-D-glucuronic acid groups. In the spectrum of Pg-xylan in Figure 4b, the presence of signals at 101.64, 78.97, 72.03, 59.42 and 57.16 ppm were attributed to the carbons of the propargyl group, indicating the successful substitution of propargyl groups on the xylan chains.

Figure 5a exhibits the ^13^C-NMR spectrum of QCS. The peaks at δ = 102.3, 62.6, 73.3, 77.9, 74.9 and 60.8 ppm were attributed to C-1, C-2, C-3, C-4, C-5 and C-6 in the chitosan unit, respectively. The peaks at δ = 51.69, 54.19, 65.33 and 69.16 ppm were attributed to C-a, C-d, C-b and C-c of the quaternary ammonium group, suggesting that quaternary ammonium groups were successfully introduced onto the chitosan backbone [40]. Figure 5b shows the ^13^C-NMR spectrum of QCS-OTs. There were strong signals between 120 and 150 ppm, attributable to tosyl groups. Moreover, the signal at 20.55 ppm originated from the methyl group. All these results indicate that the reaction between QCS and TsCl was successfully carried out. Figure 5c shows the ^13^C-NMR spectrum of QCS-N_3_; the signals originated by tosyl groups decreased, indicating that azide groups replaced tosyl groups on the QCS chains. Figure 5d shows the ^13^C NMR spectrum of xylan-click-QCS; after the Huisgen cycloaddition reaction, the signals of the triazole ring carbons were clearly identified between 120 and 140 ppm, supporting the success of the click reaction between Pg-xylan and QCS-N_3_.

### 2.2. Preparation and Characterization of Xylan-Click-QCS/AgNP Composites

#### 2.2.1. Preparation of Xylan-Click-QCS/AgNP Composites

Figure 6 shows the UV-Vis spectra of the xylan-click-QCS/AgNP composites obtained after irradiation under different reaction conditions. The typical surface plasmon resonance (SPR) band around 415 nm was observed, which indicated the formation of silver nanoparticles in the reaction. In Figure 6a, there was no SPR absorbance at 0 min, indicating that no AgNPs formed without heating. Then, the intensity of SPR peaks increased with the increase in reaction time and reached the maximum at 20 min, which confirmed the formation of more AgNPs. However, the absorption of the UV-Vis spectra increased slightly when further prolonging the heating time, indicating that an excessive reaction time has little influence on the formation of AgNPs. Thus, the optimal reaction time was determined at 20 min for this reaction.

In Figure 6b, the absorption peak was weak at 60 °C, implying that this reaction temperature was too low to generate enough AgNPs. When the reaction temperature reached 70 °C, the absorption peak was strong and quite narrow, suggesting that more AgNPs were generated, and the AgNPs were monodispersed. At 80 °C, the absorption band became weak and broadened; we speculate that a high reaction temperature may have led to the aggregation of the formed AgNPs in this reaction. Therefore, 70 °C was the appropriate reaction temperature in this experiment.

Figure 6c shows the influence of Ag^+^ contents on the formation of AgNPs. Clearly, different Ag^+^ contents had little influence on the intensity of UV-Vis spectra. However, the absorption band broadened with a slight decrease when the Ag^+^ content exceeded 0.10 mmol, and there was silver glass in the reaction, implying that excessive Ag^+^ contents may have led to the aggregation of the formed AgNPs, and the AgNPs became larger with a broader size distribution. Therefore, we chose 50 mg:0.08 mmol as the appropriate mass ratio in this experiment.

Figure 6d shows the influence of ammonia concentration on the formation of AgNPs. The absorption peak was weak when the ammonia concentration was 0.5%; then, the intensity of the absorption peaks increased with the increase in ammonia concentration. However, the ammonia concentration had little influence on absorbance intensity from 1% to 4%. Thus, 1% was chosen as the optimized ammonia concentration in this reaction.

Based on the above analysis, the best reaction conditions were as follows: under microwave irradiation, the reaction time was 20 min; the reaction temperature was 70 °C; the ratio of xylan-click-QCS to AgNO_3_ was 50 mg:0.08 mmol; and the ammonia concentration was 1%.

#### 2.2.2. Characterization of Xylan-Click-QCS/AgNP Composites

Figure 7a shows the TEM image of the AgNPs prepared in this reaction. Most of the AgNPs were spherical, but the particle size was uneven. Bigger particles with a size range of 10–20 nm are generally called Ag nanoparticles, while other smaller particles with a size range of 2~5 nm are generally called Ag nanoclusters [43]. On the one hand, this phenomenon may have been induced by the complicated component of the xylan-click-QCS composites, which may have contained unreacted xylan and QCS. On the other hand, Ag^+^ may have coordinated with the nitrogen atom of the triazole ring, which may rule out the coordination between Ag^+^ and the reducing aldehyde groups at the end of the sugar chains [36]. A longer time was needed to generate big AgNPs for the xylan-click-QCS, while the generation time was short when xylan or QCS acted as the reducing agent; the longer reaction time may have led to the aggregation of xylan- or QCS-generated AgNPs. Thus, the xylan- or QCS-generated AgNPs possessed a size in the range of 10~20 nm, while the xylan-click-QCS-generated AgNPs possessed a size in the range of 2~5 nm under these reaction conditions.

For further application of the AgNPs prepared in this reaction, the xylan-click-QCS/AgNP composites were centrifuged to obtain supernatant and sediment, respectively. The sediment dissolved in water again, and the TEM images of the sediment and supernatant AgNPs are shown in Figure 7b,c. In Figure 7b, most of the AgNPs were bigger with a size range of 5~20 nm. In Figure 7c, the Ag nanoclusters with a size range of 2~5 nm were well-dispersed in the supernatant. In addition, the high-resolution transmission electron microscopy (HRTEM) analysis of the Ag nanoclusters in the supernatant is shown in Figure 7d; the Ag nanoclusters were spherical with a clear lattice fringe, indicating the formation of Ag nanoclusters. Due to their ultra-small sizes, Ag nanoclusters possess unique physical and chemical properties compared with those of larger-sized nanoparticles, and there have been reports about Ag nanoclusters being used as electrochemical catalysts [44,45], fluorescence-based sensors [46,47], preparation materials for optical fibers [48], etc. Therefore, the prepared Ag nanoclusters could have wide applications in many areas.

### 2.3. Preparation and Characterization of Xylan-Click-QCS/AuNP Composites

#### 2.3.1. Preparation of Xylan-Click-QCS/AuNP Composites

Figure 8a–c show the UV-Vis spectra of xylan-click-QCS/AuNP composites prepared under microwave irradiation; the strong absorption peaks centered around 520 nm, relating to the characteristic surface plasma resonance (SPR) peak of AuNPs, indicating the formation of AuNPs. In Figure 8a, the absorption peak was weak at 70 °C, indicating that few AuNPs generated at this temperature. The absorbance of the UV-Vis spectra reached the maximum when the reaction temperature was 80 and 90 °C. Moreover, the UV-Vis spectrum at 90 °C was narrower with a blue shift compared with that at 80 °C, suggesting that the AuNPs became smaller with a narrow size distribution at 90 °C. However, the absorption peak became weak and broadened at 100 °C, and the solution turned to purple during the reaction, implying that part of the AuNPs aggregated under these reaction conditions. When the reaction temperature reached 100 °C, the water in the reaction system evaporated quickly, resulting in an unstable reaction system, and the generated AuNPs were uneven under this condition. Therefore, 90 °C was chosen as the optimal reaction temperature in this experiment.

In Figure 8b, the intensity of SPR peaks increased with the increase in HAuCl_4_·4H_2_O concentration and reached the maximum when the mass ratio of xylan-click-QCS to HAuCl_4_·4H_2_O was 50 mg:10 mg, which confirmed the formation of more AuNPs. Then, the absorption band became weak and broadened with a red shift when the HAuCl_4_·4H_2_O concentration further increased to 15 mg; this may have been induced by the aggregation of the AuNPs. Therefore, we chose 50 mg:10 mg as the appropriate mass ratio in this experiment. In Figure 8c, the absorbance was weak before 30 min; we speculate that this reaction may need a longer reaction time. It is worth noting that the absorbance decreased slightly when the reaction time reached 70 min. Thus, the optimal reaction time was determined to be 70 min for this reaction.

In Figure 8d, we compared the UV-Vis spectrum curves of xylan-click-QCS/AuNP composites and xylan /AuNP composites. Clearly, the AuNPs prepared by xylan-click-QCS possessed a narrow absorption band; this may be attributed to the triazole ring. Au^3+^ easily coordinated with the nitrogen atom of the triazole ring, which in turn provided a template for the nucleation and growth of the AuNPs. Moreover, the triazole ring can passivate the surface of the resultant AuNPs [49] and can finally present functional groups on the surface of the AuNPs [36].

#### 2.3.2. Characterization of Xylan-Click-QCS/AuNP Composites

Figure 9a shows the TEM image of fresh AuNPs; we can clearly see that most of the AuNPs were spherical and well-dispersed, with the diameters of 20~30 nm. Figure 9b is the high-resolution transmission electron microscopy (HRTEM) image of a single AuNP; the AuNP is spherical with a lattice fringe. In Figure 9c, the Au element in the EDS profile originated from the AuNPs reduced by xylan-click-QCS, demonstrating the formation of AuNPs. Moreover, the Cr element at 5.4 keV and Fe element at 6.4 and 6.9 keV may have originated from the specimen holder or contaminant.

### 2.4. Cell Assay

Figure 10 shows the transportation of sediment and supernatant Ag NPs during cell incubation. Obviously, sediment and supernatant Ag NPs transported into cells over a certain incubation time. It was noted that supernatant Ag NPs could be observed in the cell even after 1 h, which may be attributed to their smaller size [40]. As a result, the cells exposed to supernatant Ag NPs lost their typical shape and shrunk to a smaller size [50]. Figure 11 shows that the sediment and supernatant Ag NPs significantly inhibited the proliferation of MCF-7. Moreover, the supernatant showed stronger cell-proliferation inhibition ability than the sediment, which coincides well with the Ag NPs dispersion results during cell incubation. The result indicates that xylan-click-QCS/AgNPs could inhibit cancer.

## 3. Materials and Methods

### 3.1. Materials

Chitosan was purchased from Jinan Haidebei Marine Bioengineering Co., Ltd. (Jinan, China). Its degree of deacetylation was 85%, and its weight average molecular weight (Mw) was 2.0 × 10^5^ g/mol. Xylan isolated from bagasse was purchased from Shanghai Yuanye Biotechnology Co., Ltd. (Shanghai, China). Its Mw was 4.9 × 10^4^ g/mol. 2,3-epoxypropyltrimethyl ammoniumchloride (ETA) was purchased from Dongying Guofeng Fine Chemical Co. Ltd. (Shandong, China). Toluene sulfonyl chloride (TsCl), Propargyl bromide (PgBr) and sodium ascorbate (NaAsc) were purchased from Aladdin Reagent Co., Ltd. (Shanghai, China). Dimethyl sulfoxide (DMSO), CuSO_4_·5H_2_O and pyridine were used as received. Silver nitrate (AgNO_3_) was purchased from the Institute of Fine Chemical Material (Shanghai, China). Chloroauric acid hydrate (HAuCl_4_·4H_2_O) was purchased from ShangHai Hushi Co., Ltd. (Shanghai, China). All other chemicals were of analytical grade.

### 3.2. Synthesis and Characterization of Xylan-Click-QCS

#### 3.2.1. Synthesis of Propargyl Xylan (Pg-Xylan)

Propargyl hemicellulose was synthesized using the method by Nakagawa et al. with some modifications [51]. Briefly, 0.33 g of xylan was dissolved in 8 mL of 2% NaOH; then, we added 3 mL of H_2_O and stirred for 1 h to guarantee uniform solubilization. Subsequently, the required quantities of propargyl bromide (the molar ratios of PgBr to xylan were 0.5:1, 1:1, 2:1 and 3:1) were slowly added to the xylan solution. The reaction temperatures were subsequently kept at 30, 45, 50 and 60 °C under microwave irradiation, and the reaction was run for 30, 40, 50 and 60 min, respectively. After the required time, the product was precipitated with ethanol. Then, the precipitate was washed with ether and freeze-dried to obtain Pg-xylan. The degrees of substitution (DSs) were calculated using the following equation:(1)$$\frac{DS\times 12\times 8+\left(1-DS\right)\times 12\times 5}{12\times 5}=\frac{C\%}{C_{0}\%}\left(C_{0}=45.46\right)$$
where C is the ratio of C content in the Pg-xylan products determined by elemental analysis using an elemental analyzer (Elementar, Langenselbold, Germany), and C_0_ is the ratio of C content in xylan.

#### 3.2.2. Synthesis of Azide-6-Quaternized Chitosan (QCS-N_3_)

Synthesis of quaternized chitosan (QCS): QCS was synthesized according to the previous work [40]. Chitosan powder (1.0 g) was dissolved in acetic acid (2 wt%; 100 mL) and filtered using a 200-mesh cloth; sodium hydroxide (15 wt%) was added dropwise till complete precipitation; then, the precipitation was washed to neutral. The obtained purified chitosan was added into the mixture of isopropanol and water and then placed in a microwave reaction system. After reacting for a certain time, the reaction product was washed three times with acetone and dissolved in water. At last, the production was obtained after lyophilization at −40 °C.

The DS of quaternization groups was estimated using the deposit-titration method, which is used to determine the concentration of the chloride ion on QCS in AgNO_3_ standard solution with K_2_CrO_4_ as the indicator. The DS was calculated as follows [40]:(2)$${\mathrm{DS}}_{Q}=\frac{\frac{C_{{\mathrm{AgNO}}_{3}}\times V_{{\mathrm{AgNO}}_{3}}}{1000}}{\frac{C_{{\mathrm{AgNO}}_{3}}\times V_{{\mathrm{AgNO}}_{3}}}{1000}+\frac{W-C_{{\mathrm{AgNO}}_{3}}\times V_{{\mathrm{AgNO}}_{3}}\times M_{2}/1000}{M_{1}}}\times 100\%$$
where M_1_ stands for the glucosamine molar mass of the structural unit, g/mol; M_2_ stands for the quaternized chitosan molar mass of the structural unit, g/mol; and C_AgNO3_ (mol/L) and V_AgNO3_ (mL) represent the molarity and volume of AgNO_3_ standard solution, respectively. The DS of QCS in this study was 96%.

Synthesis of quaternized chitosan-6-p-sulfonyl ester (QCS-6-OTs): an amount of 0.5 g of chitosan was dissolved in 50 mL of H_2_O, and 1.830 g of toluene sulfonyl chloride (TsCl) was dissolved in 3 mL of pyridin. Then, we mixed the above solution and allowed it to react at room temperature for 8 h. The reaction mixture was filtered, washed with acetone and then freeze-dried to obtain QCS-6-OTs.

Synthesis of azide-6-quaternized chitosan (QCS-N_3_): QCS-6-OTs (0.5 g) was dissolved in 30 mL of H_2_O; then, the required quantities of sodium azide (the molar ratios of NaN_3_ to QCS-6-OTs were 2:1, 4:1, 6:1 and 8:1) were added to the solution. The reaction temperatures were subsequently kept at 60, 70, 80 and 90 °C under microwave irradiation, and the reaction was run for 60, 80, 100 and 1200 min, respectively. After the required time, the product was precipitated with acetone. Then, the precipitate was washed with ether and freeze-dried to obtain QCS-N_3_. The DS was calculated using the following equation:(3)$$\frac{70DS+28\times \left(1-DS\right)}{144}=\frac{N\%}{C\%}$$
where N%/C% is the ratio of N/C contents in the QCS-N_3_ products, determined by elemental analysis using an elemental analyzer (Elementar, Langenselbold, Germany), and terms 70 and 28 are the nitrogen elements in QCS-N_3_ and QCS, respectively. Term 144 is the carbon element in QCS-N_3_ and QCS.

#### 3.2.3. Synthesis of Xylan-Click-QCS Polymer via Click Chemistry

An amount of 0.3 g of Pg-Xylan was dissolved in 12 mL of DMSO, and 0.15 g of QCS-N_3_ was dissolved in 15 mL of H_2_O; then, we mixed the above solution. CuSO_4_·5H_2_O (20 mg dissolved in 1 mL of H_2_O) and sodium ascorbate (40 mg in 3 mL of H_2_O) were added. The mixture was stirred at 40 °C for 48 h and dialyzed for two days. Then, it was filtered, washed sequentially with acetone and ether and freeze-dried to obtain xylan-click-QCS.

#### 3.2.4. Characterization of the Production

Elemental analyses were carried out using an elemental analyzer (Elementar, Langenselbold, Germany). The C, H, and N contents were determined, and the substitution degree was calculated.

Fourier transform infrared (FT-IR) spectra were obtained using a Vector 33 spectrophotometer (Bruker, Bremen, Germany) in dry air at room temperature using the KBr pellets method. The spectra were collected over the range from 4000 to 400 cm^−1^ with a resolution of 4 cm^−1^.

^13^C-NMR spectra were recorded on a DRX-400 spectrometer (Bruker, Bremen, Germany) at 100 MHz, and the solvent was NaOD.

### 3.3. Preparation and Characterization of Xylan-Click-QCS/AgNP Composites

#### 3.3.1. Preparation of Xylan-Click-QCS/AgNP Composites

First, a few drops of diluted sodium hydroxide were added into some aqueous silver nitrate (1 mmol/mL). Then, different concentrations of aqueous ammonia (0.5, 1, 2, 3 and 4%) were added until all the brown silver oxide was dissolved. At this point, the mixture was clear, and aqueous silver ions existed as [Ag(NH_3_)_2_]OH in the mixture.

Xylan-click-QCS (0.05 g) was dissolved in 2% NaOH aqueous solution (5 mL) to obtain 1% xylan-click-QCS solution. The required quantities of [Ag(NH_3_)_2_]OH (the ratios of xylan-click-QCS to Ag^+^ were 50:0.05, 50:0.08, 50:0.1, 50:0.13 and 50:0.15 mg:mmol) were added to the solution. The reaction temperatures were subsequently kept at 60, 70, 80 and 90 °C under microwave irradiation, and the reaction was run for 0, 5, 10, 20, 30, 40 and 50 min, respectively.

#### 3.3.2. Characterization of Xylan-Click-QCS/AgNP Composites

UV–Vis spectra were obtained using a TU-1810 (Purkinje, Beijing, China) with a scan range of 650~250 nm.

JEM-2010HR transmission electron microscopy (TEM) (JEOL, Tokyo, Japan) was used to investigate the microstructure of the AgNPs at an accelerating voltage of 200 kV. TEM samples were prepared by diluting colloidal xylan-click-QCS/AgNP composites with water with sonication. A few drops of the suspended AgNPs were placed on a copper grid coated with an ultrathin carbon film. The particle size was measured with ImageJ software v.1.50d (NIH, Bethesda, MD, USA).

### 3.4. Preparation and Characterization of Xylan-Click-QCS/AuNP Composites

#### 3.4.1. Preparation of Xylan-Click-QCS/AuNP Composites

Xylan-click-QCS (0.05 g) was dissolved in 2% NaOH aqueous solution (5 mL) to obtain 1% xylan-click-QCS solution. The required quantities of HAuCl_4_·4H_2_O (the ratios of xylan-click-QCS to HAuCl_4_·4H_2_O were 50:2, 50:5, 50:10 and 50:15 mg:mg) were added to the solution. The reaction temperatures were subsequently kept at 70, 80, 90 and 100 °C, and the reaction was run for 5, 10, 20, 30, 40, 50, 60 and 70 min, respectively.

#### 3.4.2. Characterization of Xylan-Click-QCS/AuNP Composites

UV–Vis spectra were obtained using a TU-1810 (Purkinje, Beijing, China) with a scan range of 800~300 nm.

JEM-2010HR transmission electron microscopy (TEM) (JEOL, Tokyo, Japan) was used to investigate the microstructure of the AuNPs at an accelerating voltage of 200 kV. TEM samples were prepared by diluting colloidal xylan-click-QCS/AuNP composites with water with sonication. A few drops of the suspended AuNPs were placed on a copper grid coated with an ultrathin carbon film. The particle size was measured with ImageJ software.

### 3.5. Cell Assay

FITC (fluorescein isothiocyanate)-marked AgNP sediment and AgNP supernatant were obtained by a reaction (12 h), centrifugal and redispersion (water) process. MCF-7 cells were treated with 50 μL of FITC-marked Ag NP sediment and Ag NP supernatant for 1, 4, 12 and 24 h. The morphology of cells and Ag NP dispersion in cells were determined with a microscope (Carl Zeiss, Jena, Germany) at a 200× magnification.

Human breast adenocarcinoma cells (MCF-7) were cultured with Ag NP sediment (50 μL) and Ag NP supernatant (50 μL) for 24 h, respectively. The cell growth was evaluated with a PrestoBlue kit (Thermo Fisher Scientific, Waltham, MA, USA) according to the manufacturer’s protocol. Each test was measured in three separate wells.

## 4. Conclusions

Functional biopolymers were successfully prepared by the introduction of terminal alkyne and azide groups onto xylan and QCS chains, respectively. Subsequently, the xylan-click-QCS polymer was successfully synthesized via the Cu(I)-catalyzed click reaction. AgNPs and AuNPs were prepared by adopting xylan-click-QCS as the reducing and stabilizing agent. The prepared AuNPs with a diameter of 20~30 nm were mostly spherical and distributed uniformly. More interestingly, Ag nanoclusters generated along with AgNPs in this reaction. Ag nanoclusters could transport into cells and inhibited cell proliferation effectively. These results propose that Ag nanoclusters may be utilized as anticancer agents for the treatment of MCF-7-type cancer. The study provides a new way to combine two biopolymers for high-value applications.

## Figures and Tables

**Figure 1 molecules-27-03455-f001:**
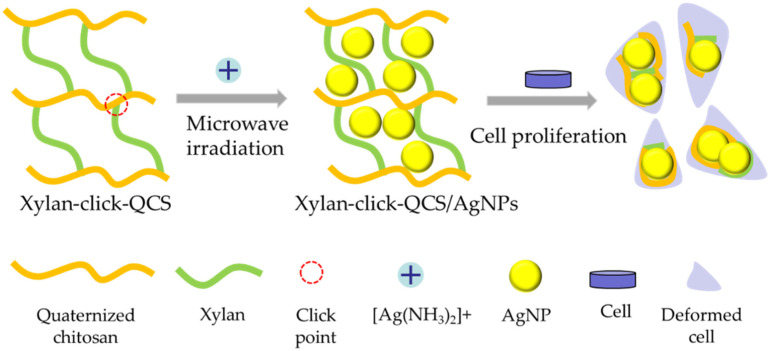
Fabrication process of xylan-click-QCS/AgNPs and its influence on cancer cells.

**Figure 2 molecules-27-03455-f002:**
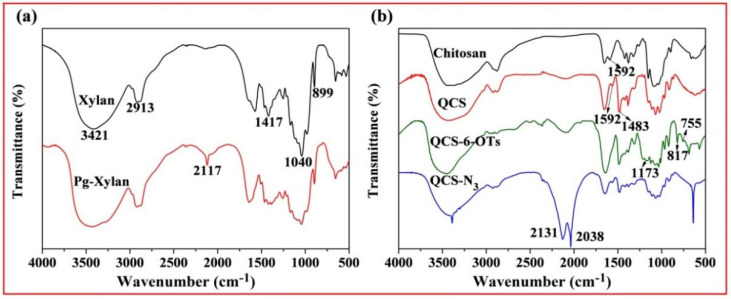
FT-IR spectra of (**a**) original xylan and Pg-xylan and (**b**) chitosan, QCS, QCS-6-OTs and QCS-N_3_.

**Figure 3 molecules-27-03455-f003:**
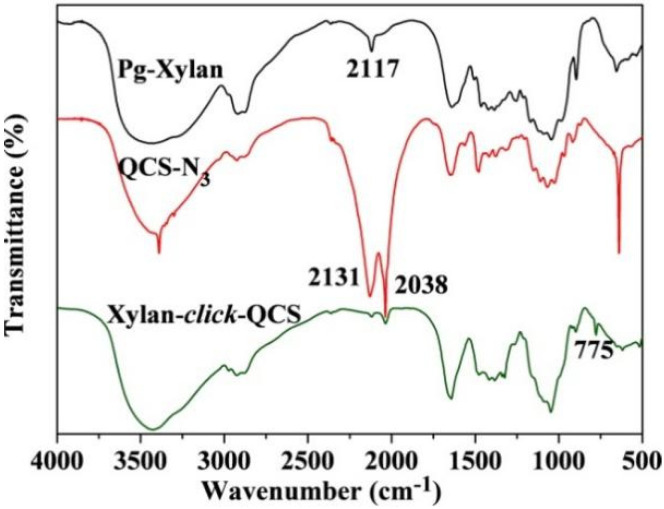
FT-IR spectra of Pg-xylan, QCS-N_3_ and Xylan-click-QCS.

**Figure 4 molecules-27-03455-f004:**
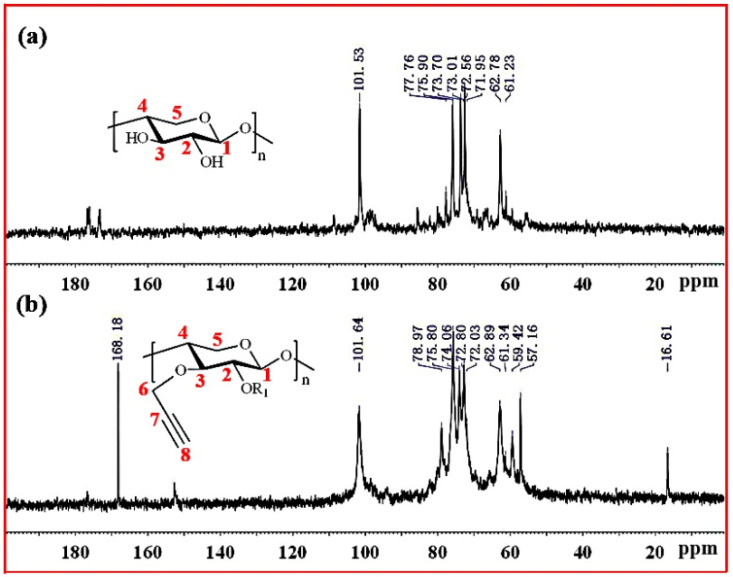
^13^C NMR spectra of (**a**) xylan and (**b**) Pg-xylan.

**Figure 5 molecules-27-03455-f005:**
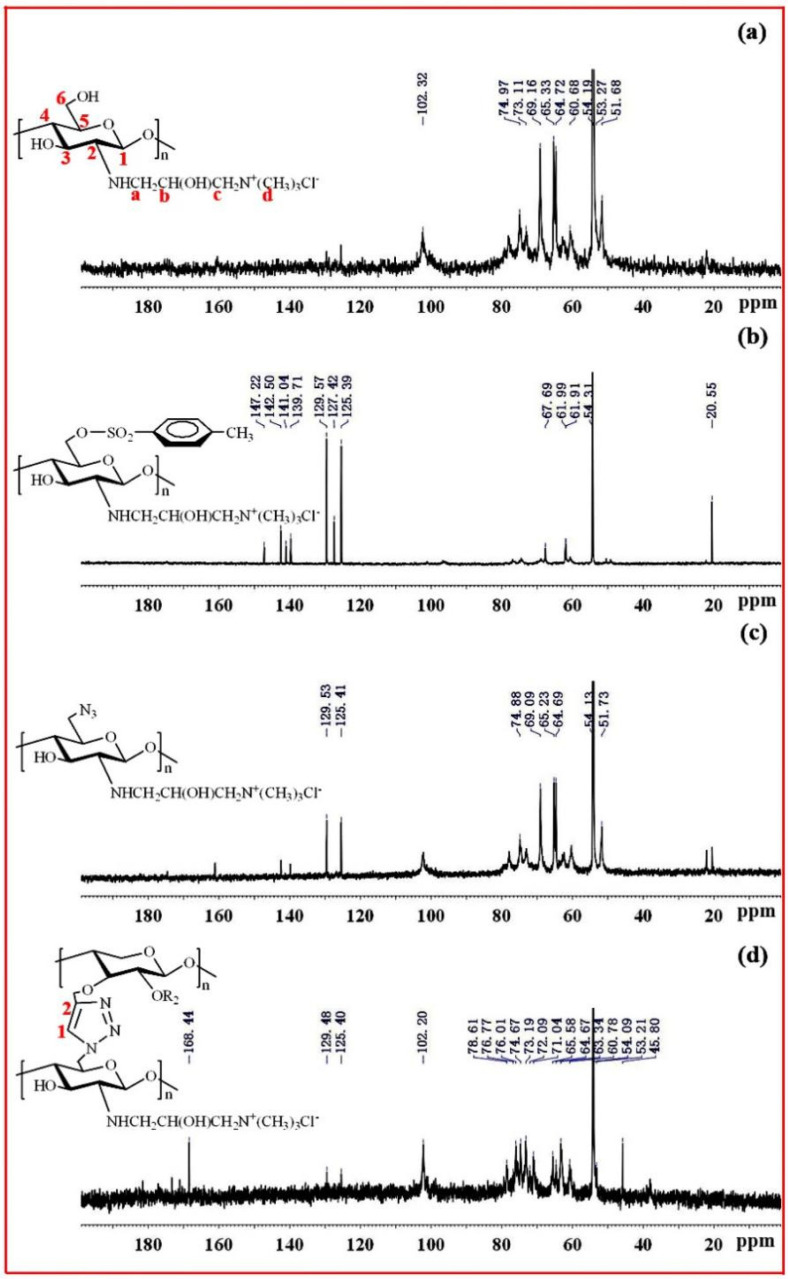
^13^C NMR spectra of (**a**) QCS, (**b**) QCS-6-OTs, (**c**) QCS-N_3_ and (**d**) Xylan-click-QCS.

**Figure 6 molecules-27-03455-f006:**
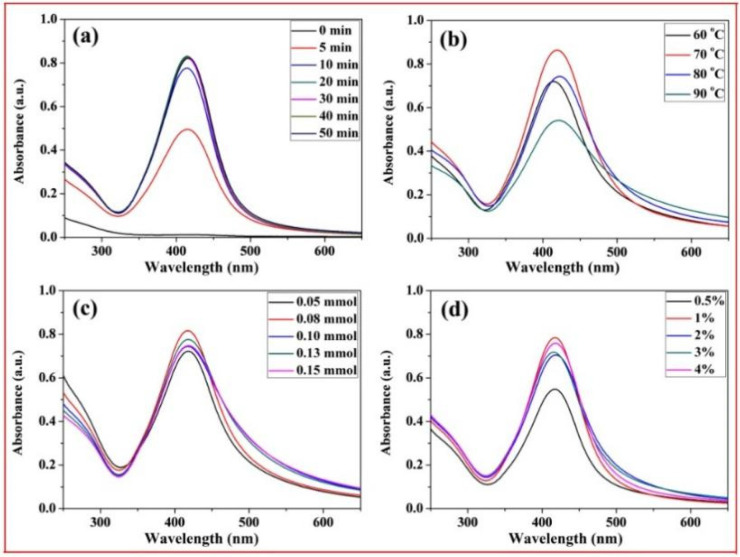
UV-Vis spectrum curves of xylan-click-QCS/AgNP composites under different reaction conditions: (**a**) different reaction time; (**b**) different reaction temperature; (**c**) different Ag^+^ content; (**d**) different ammonia concentration.

**Figure 7 molecules-27-03455-f007:**
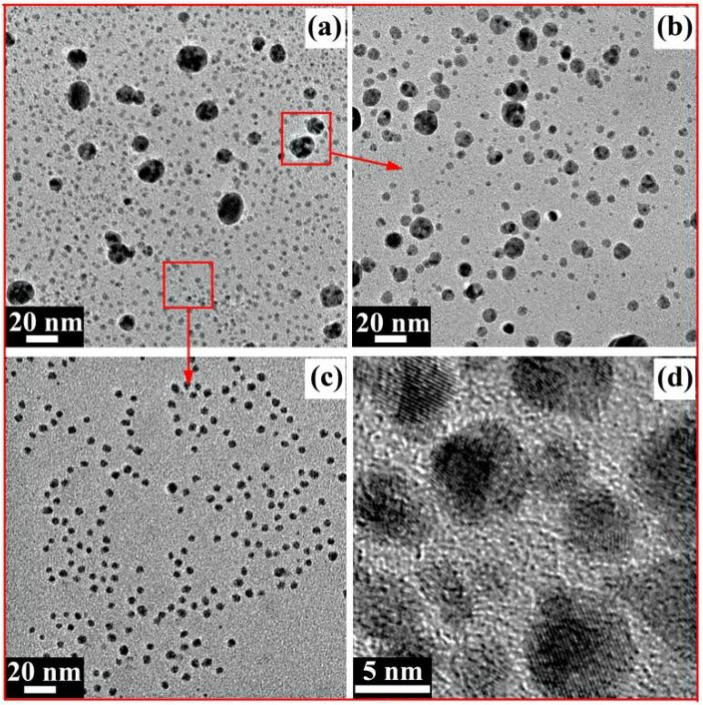
(**a**) TEM image of AgNPs; (**b**) TEM image of redispersed AgNPs in sediment; (**c**) TEM image of AgNPs in supernatant; (**d**) HRTEM image of AgNPs in supernatant.

**Figure 8 molecules-27-03455-f008:**
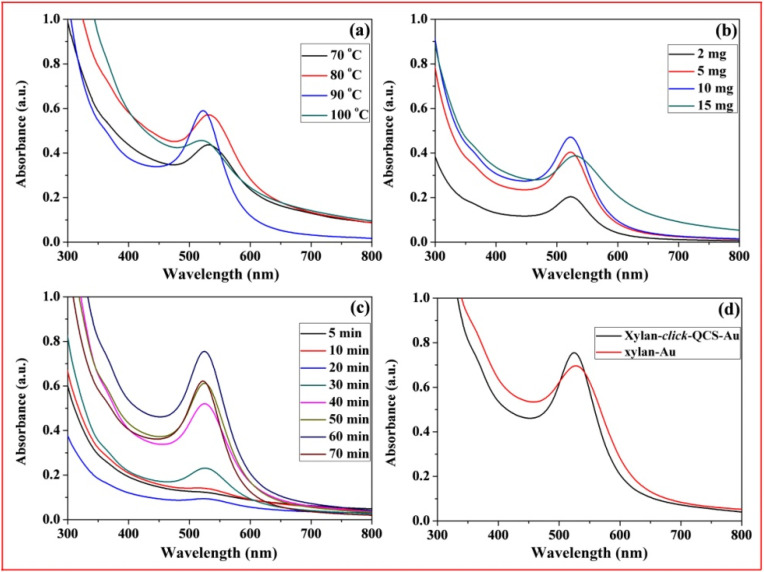
UV-Vis spectrum curves of Xylan-click-QCS/AuNP composites under different reaction conditions: (**a**) different reaction temperature; (**b**) different HAuCl_4_·4H_2_O content; (**c**) different reaction time; (**d**) comparison of UV-vis spectrum curves of Xylan-click-QCS/AuNP composites and xylan/AuNP composites.

**Figure 9 molecules-27-03455-f009:**
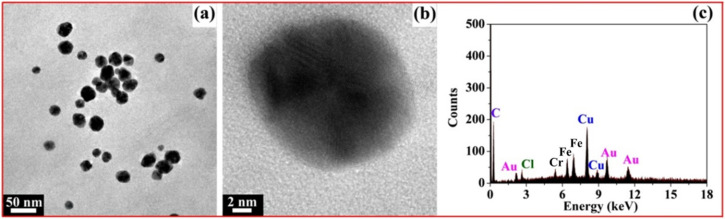
(**a**) TEM image of fresh AuNPs; (**b**) high-resolution transmission electron microscopy (HRTEM) image of single AuNP; (**c**) corresponding EDS profile of Xylan-click-QCS/AuNP composite.

**Figure 10 molecules-27-03455-f010:**
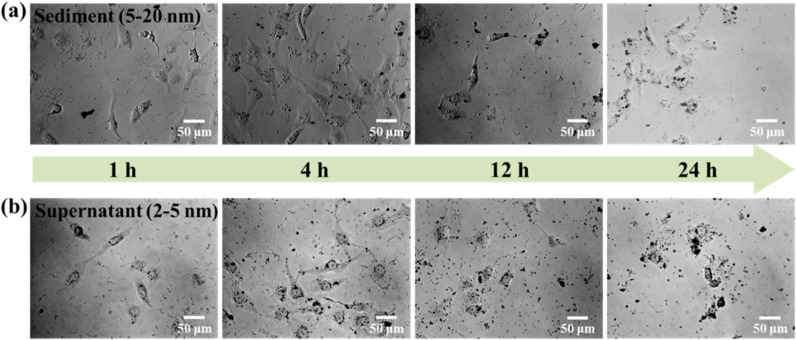
MCF-7 cells cultured with (**a**) sediment (50 μL) and (**b**) supernatant (50 μL) after 1, 4, 12 and 24 h.

**Figure 11 molecules-27-03455-f011:**
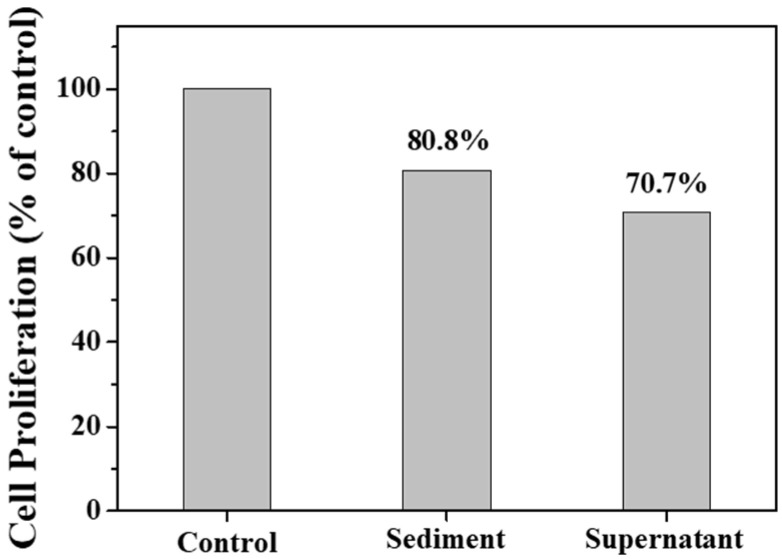
Cell proliferation assay result after 24 h of incubation with sediment and supernatant.

**Table 1 molecules-27-03455-t001:** DS of Pg-Xylan obtained under different reaction conditions.

Sample	Reaction Temperature (°C)	Reaction Time (min)	PgBr:Xylan (mol:mol)	*DS*
1	45	30	2:1	0.04
2	45	40	2:1	0.11
3	45	50	2:1	0.23
4	45	60	2:1	0.20
5	30	50	2:1	0.14
6	45	50	2:1	0.23
7	50	50	2:1	0.65
8	60	50	2:1	0.44
9	50	50	0.5:1	0.07
10	50	50	1:1	0.25
11	50	50	2:1	0.65
12	50	50	3:1	0.70

**Table 2 molecules-27-03455-t002:** DS of QCS-N_3_ obtained under different reaction conditions.

Sample	Reaction Temperature (°C)	Reaction Time (min)	NaN_3_:QCS-6-OTs (g:g)	*DS*
1	50	60	6:1	0.42
2	60	60	6:1	0.66
3	70	60	6:1	0.78
4	80	60	6:1	1.26
5	90	60	6:1	1.32
6	100	60	6:1	1.29
7	80	60	6:1	1.26
8	80	80	6:1	1.57
9	80	100	6:1	1.22
10	80	120	6:1	0.88
11	80	80	2:1	0.70
12	80	80	4:1	1.55
13	80	80	6:1	1.57
14	80	80	8:1	1.61

## Data Availability

Date are contained within the article.

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
