# Peer review of "Synthesis of Xylan-Click-Quaternized Chitosan via Click Chemistry and Its Application in the Preparation of Nanometal Materials"

_molecules, 2022, doi:10.3390/molecules27113455_

Round 1

Reviewer 1 Report

The authors investigated the preparation of xylan-click-quaternized chitosan via click chemistry and their application for the synthesis of silver nanogold nanoparticles. The manuscript has some major points that need to be corrected and explained.

What is the novel of work? Is it logical to utilize the click chemistry for the preparation of the well-known silver and gold nanoparticles? it is better to utilize such chemistry for more advanced applications. Clarify.

In lines 332 and 333, what is the concentration of the utilized acetic acid and sodium hydroxide?

The introduction part should be rewritten to include the different applications of nanoparticles using the more relevant references :

In-situ and ex-situ synthesis of poly-(imidazolium vanillyl)-grafted chitosan/silver nanobiocomposites for safe antibacterial finishing of cotton fabrics

Functionalization of cotton fabrics with titanium oxide doped silver nanoparticles: Antimicrobial and UV protection activities

Antidiabetic assessment; in vivo study of gold and core-shell silver-gold nanoparticles on streptozotocin-induced diabetic rats

Solvent-free and one-pot synthesis of silver and zinc oxide nanoparticles: activity toward cell membrane component and insulin signaling pathway in experimental diabetes

In EDX, there are some unidentified peaks. Identify them.

There are more paragraphs that have been repeated such as 33.2 and 3.4.2. Thus the manuscript should be carefully checked to avoid such mistakes and the languages need to be improved.

Particle size analyzer and zeta potential evaluation for AgNPs and AuNPs should be added.

Figure 1 should be redrawn to be more readable.

Reviewer 2 Report

The manuscript needs to be revised considering the following suggestions for further improvement:

  1. Abstract should include the results of the cell proliferation assay which author has mentioned in the methodology section (section 3.5).
  2. It is suggested to include the previous important findings of a similar line of research in the introduction section and highlight the current research gap in this area. How the present investigation is helpful to address this research gap or provide value addition in this research area should be added in the revised manuscript in the introduction section.
  3. Include an illustration that represents the overall research design of the current investigation. 
  4. The result snd discussion should be reorganized as per the heading/sub-heading of the methodology section. It will be helpful to the reader in quicker understanding.  
  5. Results of section 3.5 Cell proliferation assay is missing. It should be included in the results and discussion section of the revised manuscript.
  6.  It is advised to include the results of the cellular uptake of the developed system in the same cell line (MCF-7).
  7. Conclusion should be re-written with the inclusion of cell proliferation assay and cellular uptake study results. 

Round 2

Reviewer 1 Report

Accept as is

Reviewer 2 Report

The revised manuscript improved well and authors incorporated the given suggestions nicely. In my opinion, it should be considered for publication.